# Platelet and Lymphocyte-Related Parameters as Potential Markers of Osteoarthritis Severity: A Cross-Sectional Study

**DOI:** 10.3390/biomedicines12092052

**Published:** 2024-09-10

**Authors:** Francesca Salamanna, Stefania Pagani, Giuseppe Filardo, Deyanira Contartese, Angelo Boffa, Lucia Angelelli, Melania Maglio, Milena Fini, Stefano Zaffagnini, Gianluca Giavaresi

**Affiliations:** 1Surgical Sciences and Technologies, IRCCS Istituto Ortopedico Rizzoli, Via di Barbiano 1/10, 40136 Bologna, Italy; francesca.salamanna@ior.it (F.S.); stefania.pagani@ior.it (S.P.); melania.maglio@ior.it (M.M.); gianluca.giavaresi@ior.it (G.G.); 2Applied and Translational Research (ATR) Center, IRCCS Istituto Ortopedico Rizzoli, Via di Barbiano 1/10, 40136 Bologna, Italy; ortho@gfilardo.com (G.F.); angelo.boffa@ior.it (A.B.); lucia.angelelli@ior.it (L.A.); 3Service of Orthopaedics and Traumatology, Department of Surgery, Cantonal Hospital Authority, Via Tesserete 46, 6900 Lugano, Switzerland; 4Faculty of Biomedical Sciences, Università della Svizzera Italiana, Via La Santa 1, 6962 Lugano, Switzerland; 52nd Orthopaedic and Traumatologic Clinic, IRCCS Istituto Ortopedico Rizzoli, Via Pupilli 1, 40136 Bologna, Italy; stefano.zaffagnini@ior.it; 6Scientific Direction, IRCCS Istituto Ortopedico Rizzoli, Via di Barbiano 1/10, 40136 Bologna, Italy; milena.fini@ior.it

**Keywords:** osteoarthritis, platelet count, platelet volume, lymphocyte, platelet–lymphocyte ratio

## Abstract

Background: Platelets and lymphocytes levels are important in assessing systemic disorders, reflecting inflammatory and immune responses. This study investigated the relationship between blood parameters (platelet count (PLT), mean platelet volume (MPV), lymphocyte count (LINF), and platelet-to-lymphocyte ratio (PLR)) and osteoarthritis (OA) severity, considering age, sex, and body mass index (BMI). Methods: Patients aged ≥40 years were included in this cross-sectional study and divided into groups based on knee OA severity using the Kellgren–Lawrence (KL) grading system. A logistic regression model, adjusted for confounders, evaluated the ability of PLT, MPV, LINF, and PLR to categorize OA severity. Model performance in terms of accuracy, sensitivity, and specificity was assessed using ROC curves. Results: The study involved 245 OA patients (51.4% female, 48.6% male) aged 40–90 years, 35.9% with early OA (KL < 3) and 64.1% moderate/severe OA (KL ≥ 3). Most patients (60.8%) were aged ≥60 years, and BMI was <25 kg/m^2^ in 33.9%. The model showed that a 25-unit increase in PLR elevates the odds of higher OA levels by 1.30 times (1-unit OR = 1.011, 95% CI [1.004, 1.017], *p* < 0.005), while being ≥40 years old elevates the odds by 4.42 times (OR 4.42, 95% CI [2.46, 7.95], *p* < 0.0005). The model’s accuracy was 73.1%, with 84% sensitivity, 52% specificity, and an AUC of 0.74 (95% CI [0.675, 0.805]). Conclusions: Higher PLR increases the likelihood of moderate/severe OA, suggesting that monitoring these biomarkers could aid in early detection and management of OA severity. Further research is warranted to cross-validate these results in larger populations.

## 1. Introduction

Osteoarthritis (OA) is a chronic condition involving joint degeneration, impacting on over 300 million people worldwide [1,2,3]. It has become the third most rapidly rising condition associated with disability and given the prolonged life expectancy in many countries; the number of people affected by OA is expected to double by 2040 [2,3,4,5,6,7]. The economic burden associated with OA is also significant, contributing to increased healthcare costs and reduced productivity, particularly in aging populations. OA primarily affects the knees, hips, and hands, with the knee joint being the most commonly impacted site, leading to pain, stiffness, and functional limitations.

To date, there is considerable interest in circulating biomarkers, given the potential for early identification and treatment of patients at risk of OA prior to the development of irreversible clinical disease (Figure 1). Macrophages and pro-inflammatory cytokines like interleukin (IL)-1, IL-6, IL-8, and tumor necrosis factor (TNF)-α are key components in the process [8]. These inflammatory mediators not only contribute to cartilage degradation but also promote synovial inflammation and osteophyte formation, further exacerbating joint damage. Additionally, the role of oxidative stress and mechanical stress in OA pathogenesis has gained attention, as these factors may influence the release of biomarkers associated with disease progression. C-terminal telopeptides of type I collagen (CTX I), CTX II, type III collagen N-propeptide, cartilage oligometric matrix protein (COMP), interferon-γ inducible protein 10 (IF-γ IP-10), matrix metalloproteinase-3 (MMP3), MMP-2, adiponectin, ILs, and ghrelin were suggested as potential OA biomarkers [7,8], but potential biomarkers can belong to a broad range of categories. A systematic review by Boffa et al. describes a “biomarker” as a characteristic that is measured and assessed as an indicator of normal biological processes, pathogenic processes, or pharmacologic responses to therapeutic intervention, potentially represented by radiographic, histologic, physiologic, or molecular characteristics [9]. In addition to these characteristics, an ideal OA biomarker should also be simple, safe, sustainable, of high-quality, and patient-friendly with a maximum health care impact. Moreover, the biomarker should ideally correlate with clinical symptoms such as pain and functional impairment, providing a more comprehensive understanding of the disease state. In this context, several parameters of complete blood cell count (CBC) examination, routinely used for the systemic evaluation, were recently explored as prognostic and/or diagnostic biomarkers in various diseases, including OA [10,11,12,13,14,15,16,17].

In OA, these biomarkers are based on parameters related to platelet (PLT) count, mean platelet volume (MPV), lymphocyte (LINF) count, and platelet–lymphocyte ratio (PLR) [13,14,18,19,20,21,22]. With reference to these parameters, some studies demonstrated that MPV and PLR are independent predictors of severe hip OA [14,18,19] and that high PLT counts within the normal range are significantly associated with knee and hip OA in Korean women over 50 years old [20]. These findings suggest that platelet function and inflammation play a crucial role in the pathophysiology of OA, potentially offering new avenues for therapeutic intervention. An increase in PLT count is related to the reactivity of megakaryocytes, and to the activity of released pro-platelet cytokines. This inflammatory response may contribute to the chronic nature of OA, perpetuating joint damage over time. This is hardly surprising as platelets can also be used as a treatment approach for OA [23]. In fact, multiple randomized controlled trials, systematic reviews, and meta-analyses prove the effectiveness of platelet-rich plasma (PRP) as a minimally invasive method of pain alleviation in OA treatment [24,25,26,27]. PRP therapy involves the concentration of autologous platelets in a small volume of plasma, which is then injected into the affected joint to promote tissue repair and reduce inflammation. Additionally, it was also reported that neutrophil–lymphocyte ratio and neutrophil–monocyte ratio values were higher in patients with hip OA than in healthy controls [19]. These hematological markers may reflect systemic inflammation, which is increasingly recognized as a contributor to OA pathogenesis.

However, exploration and interpretation of OA-related alterations of PLT and LINF counts, MPV, and PLR should also be evaluated considering age, gender, and body mass index (BMI). In fact, a critical effect of aging influences the counts of cytotoxic and helper T cells, a specific type of LINF [28]. During the human lifetime, the percentages of naive T cells within both CD4+ and CD8+ cells decline considerably from 45% and 43% to 20% and 10%, respectively [28]. Differently, the proportion of central memory and effector memory cells within cytotoxic and helper T LINF increases with age [28]. In addition to age, gender also influences LINF counts [29]; in fact, counts of total CD3+ T LINFs as well as CD8+ cytotoxic T cells and especially CD4+ T helper cells are higher in females compared to males [29]. These gender differences might be influenced by hormonal factors, particularly estrogen, which has been shown to modulate immune function and inflammatory responses. Similarly, age can also influence PLT count [30], as in newborns PLT counts are typically higher, often around 150,000 to 400,000 PLTs per microliter [30], while this count gradually decreases as a person ages [31]. As with age, PLT count can also vary between genders: typically, males tend to have slightly higher PLT counts compared to females [30]. Another PLT parameter that correlates with PLT counts is the MPV [32,33], although this correlation is nonlinear [32]. In fact, unlike PLT count, MPV is genetically determined and is not affected by gender or lifestyle, while the influence of age on MPV is still debated [31,33]. This complexity underscores the need for further research to clarify the relationship between MPV and OA, particularly in different demographic groups. Regarding BMI, there is no direct correlation between BMI and blood parameters. However, some research suggests that obesity (BMI ≥ 30) can lead to chronic low-grade inflammation, which may affect various aspects of blood, including PLT and LINF count and function [34,35,36,37]. However, the clinical significance of these differences is still under investigation. Obesity is a well-known risk factor for OA, primarily due to the increased mechanical load on weight-bearing joints, but its role in modulating systemic inflammation and blood parameters remains a key area of interest.

Considering these data, the question arises whether the reported alterations in PLT and LINF counts, MPV and PLR in patients with OA are indeed primarily due to OA, or whether gender, age, and BMI might have a significant impact. This distinction is crucial for the accurate interpretation of biomarkers and their potential use in personalized medicine. Given the absence of research investigating these key aspects, the present cross-sectional study aimed to examine specific blood platelet parameters, such as PLT and LINF counts, MPV and PLR as OA biomarkers, stratifying for OA grade, age, gender, and BMI differences.

## 2. Materials and Methods

### 2.1. Study Population

This cross-sectional study was approved by the Ethics Committee of (blinded for review) (approval number: CE-AVEC 582/2021/Oss/’blinded for review’) and performed in accordance with the Helsinki Declaration. All patients signed informed consent forms to allow access to their medical records. Clinical, radiographic, and biological data in a pseudo-anonymous form from patients affected by various degrees of knee OA (Kellgren–Lawrence (KL) grading 1–4) and treated at the (blinded for review) from 2009 to 2021 were collected via their electronic medical records (EMR). The inclusion criteria were patients aged >40 years, of both male and female genders, who were treated for knee OA. Additionally, eligible patients underwent X-ray evaluation for the OA degree and had a complete blood count including PLT, LINF, MPV, and C-reactive protein (CRP). The study excluded patients with a history of hematological and/or coagulation disorders and/or previous thromboembolic events, uncompensated systemic inflammatory diseases, rheumatic pathologies, post-traumatic OA, and oncological diseases. Furthermore, patients who had undergone surgery in the previous 6 months and patients who did not give consent to access their medical records were also excluded.

Two hundred and forty-five patients were included in the final analysis. Two physicians reviewed patients EMR and extracted the following data: age, sex, height, weight, BMI, KL grading, PLT and LINF count, MPV and PLR values.

### 2.2. Radiologic Assessment

The KL grading system was used for classifying radiographic OA by two physicians, with disagreements resolved by consensus with a third senior physician. Briefly, the system uses four radiographic features: joint space narrowing, osteophytes, subchondral sclerosis, and subchondral cysts. The severity of radiographic changes increases from grade 0 to 4 with grade 0 meaning no radiographic features of OA whereas grade 4 meaning large osteophytes, marked joint space narrowing, severe sclerosis, and definite bony deformity [38]. If noticeable KL grading differences were present between the two knees, the grading of the most severely affected one was recorded.

Disagreements between the two initial physicians regarding radiographic OA classification were typically related to borderline cases where joint space narrowing, or osteophyte formation, was subtle. These disagreements were resolved through a consensus process involving a third senior physician. The three physicians reviewed the radiographic images together and discussed the specific features and grading criteria until an agreement was reached. This collaborative approach helped ensure consistency in the KL grading and minimized subjectivity.

### 2.3. Laboratory Parameters

Peripheral venous blood samples were collected using standard medical procedures. Complete blood count parameters were analyzed by (blinded for review) and PLT and LINF count, MPV and PLR values were extracted and analyzed.

### 2.4. Statistical Analysis

All statistical analyses were conducted using R Statistical Software v.4.4.0 [39] and specific additional packages such as pROC v.1.18.5 [40]. The data were presented as mean values and 95% confidence intervals (95% CI) for continuous variables, and absolute frequencies and percentages for categorical variables. The significance of differences between groups was evaluated using the chi-square or Fisher exact test for categorical variables, or the *t*-test or Wilcoxon–Mann–Whitney test for continuous variables, as appropriate. A logistic regression model (generalized linear model for binomial family), adjusted for confounding variables, was constructed to assess the ability of the tested blood parameters to classify patients into early and moderate/severe OA and calculate their related odds ratio (OR) and 95% confidence intervals (95% CI) in patients with moderate/severe OA compared to those with early OA. Before fine-tuning the logistic regression model, it was checked that there was no multicollinearity, i.e., that the independent variables were not correlated with each other (Kendall correlation), and that the continuous independent variables had a linear relationship with the log odds of the dependent variable (Box–Tidwell test). A k-fold cross validation was employed to ensure that the results were robust and generalizable across different subsets of the data and to detect overfitting (k = 20). The performance of the selected logistic regression model was then evaluated using a receiver operating characteristic (ROC) curve. This allowed the accuracy, sensitivity, and specificity of the model in classifying OA severity to be defined.

## 3. Results

The demographic and clinical parameters of the patients are presented in Table 1. A total of 245 participants with OA, aged between 40 and 90 years, were included in the study. Early OA (KL < 3) was present in 88 (35.9%) patients, while 157 (64.1%) had moderate/severe OA (KL ≥ 3). The gender distribution was 51.4% female and 48.6% male. A total of 39.2% of the patients were aged between 40 and 60 years, while the remaining 60.8% were aged ≥60 years. The BMI of the participants was less than 25 kg/m^2^ in 33.9% of cases and between 25 and 30 kg/m^2^ in 66.1%.

Table 2 presents the results of the hematologic parameters considered, stratified by OA KL grade (<3 and ≥3), as well as by sex, age, and BMI in the selected classes.

A binary logistic regression model was employed to assess the influence of several independent variables, including age, sex, BMI, and the counts of LINF and PLT, MPV, and PLR, on the probability of OA level detection. The dependent variable was the occurrence of OA detection from early to moderate/severe grade (coded as 0 and 1, respectively). As the number of patients with OA Grade ≥ 3 increased, we predicted that these patients were experiencing higher OA Grade value; the accuracy of the baseline model is 64.1%. To understand better the variations in hematologic parameters in relation to the severity of OA, the data were analyzed using boxplots, as shown in Figure 2. The boxplots represent the distribution of PLT, MPV, LINF, and PLR among patients with early OA (KL < 3) and those with moderate/severe OA (KL ≥ 3).

Prior to defining the model, it was confirmed that there was no multicollinearity, indicating that there was no correlation between the independent variables, except between PLT and MPV (Kendall’s τ = −0.328, *p* = 0.041) and between LINF and PLR (Kendall’s τ = −0.574, *p* < 0.005). Subsequently, the Box–Tidwell test was employed to validate the assumption of linearity in the logarithm of probabilities for the continuous independent variables. This assumption was not validated for LINF, with a *p* < 0.0005. Univariate regression logistic models showed that LINF (OR = 0.63, 95% CI [0.42, 0.94], *p* = 0.023) and PLR (OR = 1.008, 95% CI [1.003, 1.013], *p* = 0.0014) might be considered as potential predicting factors of OA higher level, but LINF as a protective factor.

The selected model (PLR, age in classes and sex as independent variables) demonstrated a satisfactory fit (PLR = 44.8, *p* < 0.0005, AIC = 274.83, and the Hosmer–Lemeshow test *p* = 0.345) and the pseudo Nagelkerke R^2^ value was 0.234. Furthermore, the variance inflation factor was found to be less than 1.5, confirming the absence of multicollinearity among the independent variables. Age classes (<40 and ≥40 years old) and sex were inserted in the model to adjust the coefficient regression of the continuous variable PLR, but it had been previously investigated that they did not have a confounding role (Wald tests among strata were not significant).

A 25-unit increase in PLR level elevates the odds of higher OA level by 1.30 times (1-unit OR = 1.011, 95% CI [1.004, 1.017], *p* < 0.005), while being aged ≥ 40 years elevates the odds by 4.42 times (OR 4.42, 95% CI [2.46, 7.95], *p* < 0.0005). The observed 38% higher odds ratio for males compared to females (OR 1.38, 95% CI [0.77–2.48], *p* = 0.279) was not statistically significant. The ROC curve analysis provided further validation of the model’s predictive power (Figure 3). The classification accuracy of the model was 0.739 (95% CI [0.678, 0.793]) which was significantly better than the no-information rate (*p* = 0.002) with a sensitivity of 84% and a specificity of 52%. The area under the curve (AUC) was 0.74 (95% CI [0.675, 0.805]), indicating that the model has good discriminatory ability to distinguish between moderate/severe and early OA classes.

## 4. Discussion

OA is a degenerative joint disease determined by the imbalance of pro-inflammatory and anti-inflammatory mediators such as IL-1α, IL-1β, IL-4, IL-6, IL-8, IL-10, IL-11, IL-13, IL-15, IL-17, TNF-α, leukocyte inhibitory factor, IL-1 receptor antagonist, matrix metalloproteinases, proteases, chemokines, nitric oxide, prostaglandins, and leukotrienes [7,8]. These mediators are primarily produced by synoviocytes and chondrocytes in the affected joints and are released into the synovial fluid [3,7,8]. Furthermore, IL-1, IL-6, IL-15, and TNF-α are also detectable in the serum of patients with OA [7,8]. Several inflammatory mediators, particularly IL-6, stimulate PLT production from megakaryocytes [19]. This inflammatory state leads to PLT activation and prolonged survival, which in turn triggers leukocyte activation, creating a feedback loop. Consequently, the PLT count in peripheral blood increases. In fact, some studies have highlighted significant alterations in PLT parameters and LINF among OA patients [19,20,21]. In addition, research has also demonstrated that elevated PLT counts are correlated with increased joint inflammation and pain severity in OA patients [19,20,21]. However, conflictingly, Korkmaz et al. found no relationship between PLR and the severity of hip OA in a recent retrospective study [25]. Notably, the mean age of participants in this study was lower than that in the previously cited research. It is well-established that the inflammatory process tends to increase with age. This has been supported by other studies showing that older individuals with OA often exhibit more pronounced systemic inflammation [19,20,21,25]. The significance or lack thereof in PLR values could potentially be attributed to the younger age of participants or other demographic factors such as gender and BMI. Thus, our investigation aimed to explore the relationship between PLT, MPV, LINF, and PLR with OA severity while also considering demographic confounding variables including age, sex, and BMI.

In this study on the analysis of a cohort of 245 knee OA patients, the logistic regression model identified several key predictors of the OA level. Higher PLR levels and older age were significantly associated with an increased risk of progressing to severe OA. Specifically, a 25-unit increase in PLR was associated with a 1.30-fold increase in the odds of having more severe OA, while being aged ≥40 years elevated the odds by 4.42 times. This aligns with findings from other studies suggesting that higher systemic inflammation markers are linked with more severe OA outcomes [41,42,43,44,45]. These results underscore the importance of systemic inflammatory markers and age in the pathophysiology of OA. In fact, both epidemiological and biological evidence suggest a connection between age-related inflammation, also known as ‘inflammaging’, and the onset of OA [41,42]. Age-related pro-inflammatory mediators contributing to OA may stem from peripheral sources, such as adipose tissue, which tends to increase with age, as well as from local production within joint tissues [43,44]. While cell senescence and the development of the senescence-associated secretory phenotype offer a compelling mechanism linking aging, inflammation, and OA, there is insufficient evidence to suggest that this occurs with normal aging in joint tissues [45]. Excessive mechanical loading of the joint, severe enough to induce OA, may lead to stress-induced senescence and heightened production of pro-inflammatory mediators; however, whether this phenotype occurs in normally aged joints not subjected to excessive loading remains less clear [46,47]. Recent studies have shown that both mechanical stress and intrinsic aging contribute to joint inflammation and cartilage degradation, but their relative impacts may vary depending on the specific OA context [48].

Regarding potential gender differences, in this study males exhibited a 38% higher odds ratio of having moderate/severe OA compared to females; however, this difference did not reach statistical significance. This suggests that gender differences in OA progression trends may not be statistically significant within the scope of this study. Historically, OA has been perceived as more prevalent in women, particularly in older age groups [49]. However, recent studies have challenged this notion, indicating that OA may affect men and women equally, albeit with variations in joint distribution [49,50]. For instance, studies have found that men may present with OA at a younger age compared to women, possibly due to differing patterns of joint use and biomechanical stress [49,50]. These differences in joint involvement may reflect underlying anatomical and biomechanical differences between genders [50]. Furthermore, while gender differences in OA progression are less well-defined, some studies suggest that men may experience a more rapid progression of joint damage and functional decline compared to women [50]. This rapid progression in men might be influenced by higher levels of physical activity or different injury mechanisms. While gender differences in OA are increasingly recognized, their precise nature and underlying mechanisms remain complex and multifaceted. Future research should aim to unravel the interplay of gender-specific factors contributing to OA severity and progression, such as hormonal influences, biomechanical differences, and socio-cultural factors.

Similarly, also BMI did not show any significant differences in PLT parameters and LINF levels when considered within the two main ranges (BMI < 25 kg/m^2^ and 25–30 kg/m^2^). This indicates that, within these limits, BMI is not a determining factor for blood alterations in OA. However, this finding is consistent with other research suggesting that moderate ranges of BMI may not significantly affect these hematologic parameters [51]. However, the absence of patients with BMI > 30 kg/m^2^ in the study limits our understanding of the impact of severe obesity. Given the known association between obesity and systemic inflammation, future research should aim to include a broader range of BMI categories to fully elucidate the relationship between obesity and OA severity [46]. Studies have consistently shown that extreme obesity is associated with higher levels of systemic inflammation and may exacerbate OA symptoms [46,51].

Finally, we also constructed a final regression model aimed at demonstrating the predictive power of this analysis. Through rigorous analysis and validation, we aimed to showcase the model’s ability to predict accurately outcomes and provide valuable insights for decision-making and future research endeavors. The model demonstrated good predictive power, with a classification accuracy of 73.1%, a sensitivity of 84%, and a specificity of 52%. The area under the ROC curve (AUC) was 0.74, indicating a good discriminatory ability to distinguish between early and moderate/severe OA classes. This suggests that the model can effectively differentiate between stages of OA and potentially guide clinical decision-making. The performance metrics presented suggest that the developed regression model has promising predictive capabilities for assessing OA severity based on demographic and hematologic parameters. The combination of reasonable classification accuracy, high sensitivity, and acceptable AUC indicates that the model could be a valuable tool in clinical practice.

Despite the strengths of this study, including a large, well-defined cohort and robust statistical analyses, there are some limitations to consider. The cross-sectional design precludes establishing causality, and longitudinal studies are needed to confirm these findings. Additionally, while the model accounted for several key variables, other potential confounders, such as physical activity, diet, and genetic factors, were not included and should be explored in future research. Finally, considering the positive correlation between platelet size, distribution width, volume changes, and low bone mineral density due to osteoporosis, future investigations will also consider how the coexistence of OA and osteoporosis may influence platelet parameters and lymphocytes [52]. To address the limitations of our study, future research could implement longitudinal designs to establish causal relationships and validate our findings over time. Additionally, incorporating a broader range of potential confounders such as physical activity, diet, and genetic factors would enhance the accuracy and applicability of the model. Exploring the interplay between osteoarthritis and osteoporosis, particularly how their coexistence affects platelet parameters and lymphocytes, could offer deeper insights into their combined effects on hematologic parameters and provide a more comprehensive understanding of these conditions.

## 5. Conclusions

In our opinion, this study underscores the valuable role of inflammatory markers like PLR in predicting the severity of OA. We believe that incorporating these biomarkers into clinical practice could enhance early detection and individualized treatment plans. While our findings did not reveal significant effects of gender or BMI on OA severity, we acknowledge the need for further research to explore these factors more thoroughly. We are confident that ongoing research into the inflammatory pathways of OA will lead to more effective management strategies and improved patient outcomes.

In conclusion, this study provides significant insights into biomarkers related to OA severity. The identification of PLR and age as significant predictors of OA severity has important clinical implications. Monitoring these parameters in patients with OA could aid in the early identification of individuals at higher risk of disease worsening. This could facilitate timely interventions aimed at mitigating disease progression, improving patient outcomes, and reducing the burden of OA on healthcare systems, supporting population stratification to optimize access to the most invasive and expensive tests. Moreover, early identification based on these biomarkers could lead to more personalized treatment strategies. However, future studies should aim to validate these findings in larger, more diverse cohorts and explore additional factors that may identify OA severity.

Bullet points:This study provides valuable information on biomarkers related to osteoarthritis (OA) severity.The identification of PLR and age as key predictors of OA severity has important clinical implications.Monitoring these parameters in OA patients could help in the early identification of individuals at higher risk of disease worsening.This could enable timely interventions to mitigate disease progression, improve patient outcomes, and reduce the burden of OA on healthcare systems.Population stratification could optimize access to the most invasive and expensive tests.Early identification based on these biomarkers could lead to more personalized treatment strategies.Future studies should aim to validate these findings in larger, more diverse cohorts and explore additional factors that may identify OA severity.

## Figures and Tables

**Figure 1 biomedicines-12-02052-f001:**
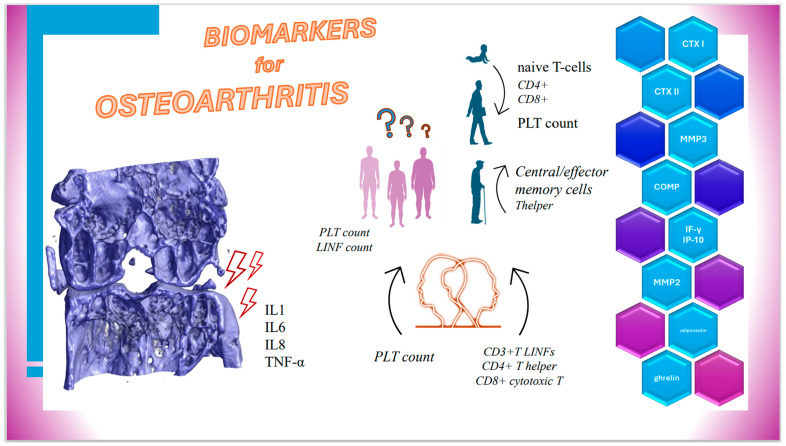
Schematic representation of the biomarkers in osteoarthritis. IL: interleukin; TNF-α: tumor necrosis factor-α; PLT: platelet; LINF: lymphocyte; CTX I and II: C-terminal telopeptides of type I and II collagen; MMP: matrix metalloproteinase; COMP: cartilage oligometric matrix protein; IF-γ IP-10: interferon-γ inducible protein 10.

**Figure 2 biomedicines-12-02052-f002:**
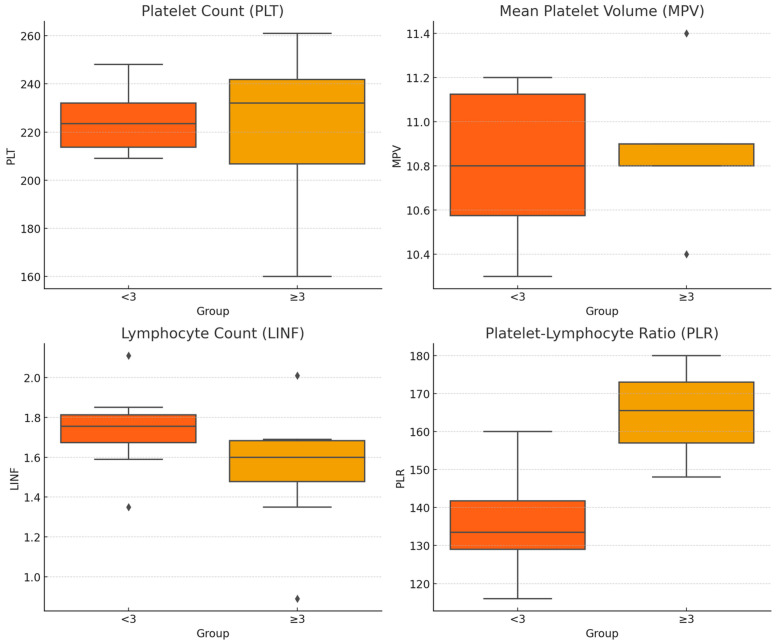
Boxplot of hematologic parameters by KL grade of OA. The data show significant variations in PLT levels and differences in the distribution of MPV and LINF between KL grades.

**Figure 3 biomedicines-12-02052-f003:**
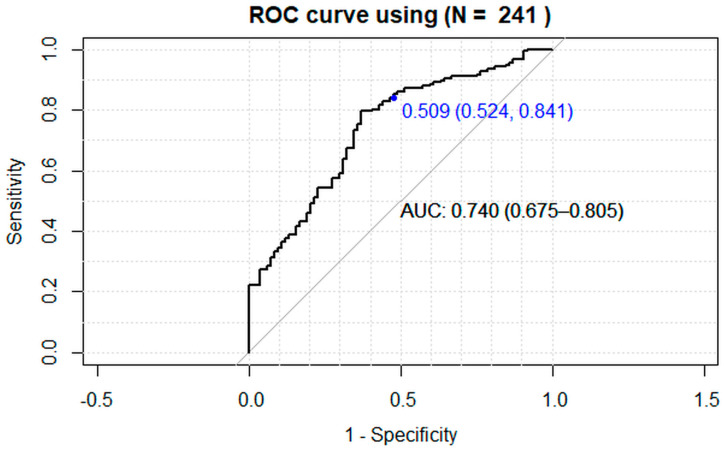
ROC curve analysis.

**Table 1 biomedicines-12-02052-t001:** Demographic characteristics and clinical characteristics of OA patients according to KL score severity. Data are reported as N or mean [95% CI].

		Patients	KL < 3	KL ≥ 3	*p*-Value
	N	245	88	157	
Sex	F	126	50	76	0.232 *
	M	119	38	81	
Age	yrs	65 [64, 66]	59 [58, 61]	68 [66, 69]	<0.0005 °
	40–60	96	53	43	<0.0005 *
	>60	149	35	114	
BMI	kg/m^2^	25.8 [25.6, 26.1]	25.6 [25.2, 26.0]	26.0 [25.6, 26.3]	0.2879 °
	<25 kg/m^2^	83	32	51	0.575 *
	25–30 kg/m^2^	162	56	106	

KL: Kellgren–Lawrence, BMI: body mass index, F: female, M: male, yrs: years. * Fisher exact test; ° Student *t*-test.

**Table 2 biomedicines-12-02052-t002:** Results of hematologic parameters stratified for OA KL grade, sex, age, and BMI selected classes. Mean [95% CI].

OA_KL Grade	Sex	Age	BMI	N	PLT 10^9^/L	MPV fL	LINF 10^9^/L	PLR 10^9^/L
<3	F	40–60	<25	15	215 [205, 225]	11.2 [11.0, 11.4]	1.70 [1.60, 1.80]	135 [125, 145]
			25–30	13	232 [213, 251]	11.2 [10.9, 11.5]	1.85 [1.77, 1.93]	129 [116, 142]
		>60	<25	5	216 [201, 231]	10.7 [10.4, 11.0]	1.59 [1.50, 1.68]	137 [126, 148]
			25–30	17	248 [232, 264]	10.9 [10.7, 11.1]	1.71 [1.56, 1.86]	160 [145, 175]
	M	40–60	<25	9	231 [223, 239]	10.6 [10.3, 10.9]	1.80 [1.67, 1.93]	132 [123, 141]
			25–30	16	232 [218, 246]	11.1 [10.9, 11.3]	2.11 [1.96, 2.26]	116 [107, 125]
		>60	<25	3	209 [138, 280]	10.3 [9.6, 11.0]	1.35 [1.00, 1.70]	156 [127, 185]
			25–30	10	210 [193, 227]	10.5 [10.2, 10.8]	1.80 [1.59, 2.01]	129 [114, 144]
≥3	F	40–60	<25	11	261 [237, 285]	10.9 [10.7, 11.1]	2.01 [1.74, 2.28]	148 [128, 168]
			25–30	13	250 [235, 265]	10.9 [10.7, 11.1]	1.69 [1.56, 1.82]	154 [144, 164]
		>60	<25	18	236 [221, 251]	10.8 [10.6, 11.0]	1.59 [1.45, 1.73]	173 [153, 193]
			25–30	34	239 [226, 252]	10.8 [10.6, 11.0]	1.68 [1.59, 1.77]	158 [146, 170]
	M	40–60	<25	2	160 [128, 192]	11.4 [10.7, 12.1]	0.89 [0.71, 1.07]	180 [152, 208]
			25–30	17	206 [195, 217]	10.9 [10.7, 11.1]	1.35 [1.22, 1.48]	173 [158, 188]
		>60	<25	20	228 [216, 240]	10.4 [9.9, 10.9]	1.61 [1.41, 1.81]	168 [153, 183]
			25–30	42	207 [198, 216]	10.9 [10.8, 11.0]	1.52 [1.40, 1.64]	163 [151, 175]

## Data Availability

Data will be shared upon request to the corresponding author.

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
