# Peer review of "Platelet and Lymphocyte-Related Parameters as Potential Markers of Osteoarthritis Severity: A Cross-Sectional Study"

_biomedicines, 2024, doi:10.3390/biomedicines12092052_

Round 1
Reviewer 1 Report
Comments and Suggestions for Authors
This manuscript "PLATELET AND LYMPHOCYTE-RELATED PARAMETERS AS POTENTIAL MARKERS OF OSTEOARTHRITIS SEVERITY: A CROSS-SECTIONAL STUDY" investigated the relationship between blood parameters (platelet count [PLT], mean platelet volume [MPV], lymphocyte count [LINF], and platelet-to-lymphocyte ratio [PLR]) and osteoarthritis (OA) severity, considering age, sex, and body mass index (BMI) to examine the specific blood platelet parameters as OA biomarkers, stratifying for OA grade, age, gender, and BMI differences. Overall, this article is good and covers important area. These are the suggested corrections:
1- The introduction feels a bit basic.
2- The presentation of table 2 needs to be improved because I feel that there is overlapping between the data.
3- I suggest the authors use charts to show the influence of the investigated parameters.
4- The discussion should be improved; there are a few paragraphs that compare the data acquired with what has previously been reported in the literature.
5- Authors perspectives and opinions need to be included in the conclusion part.
Comments on the Quality of English LanguageMinor editing.
Author Response
1- The introduction feels a bit basic.
We thank the reviewer, and following the suggestions received, we have modified and enhanced the introduction to make it less basic and more informative. We have highlighted all the changes we made in red.
2- The presentation of table 2 needs to be improved because I feel that there is overlapping between the data.
We have tried to make the table easier to understand by adding rows and columns and expanding the spacing.
3- I suggest the authors use charts to show the influence of the investigated parameters.
As suggested, to better understand the variations in hematologic parameters in relation to the severity of osteoarthritis, the data were analyzed using boxplots. These boxplots represent the distribution of PLT, MPV, LINF, and PLR among patients with early OA (KL < 3) and those with moderate/severe OA (KL ≥ 3).
4- The discussion should be improved; there are a few paragraphs that compare the data acquired with what has previously been reported in the literature.
As requested by the reviewer, we have enhanced the discussion by more thoroughly incorporating and comparing our results with those currently available in the literature.
5- Authors perspectives and opinions need to be included in the conclusion part.
As suggested we added authors perspectives and opinions in the conclusion part.
Comments on the Quality of English Language
Minor editing.
We have reviewed the English in the text.
Reviewer 2 Report
Comments and Suggestions for Authors
The manuscript "PLATELET AND LYMPHOCYTE-RELATED PARAMETERS AS POTENTIAL MARKERS OF OSTEOARTHRITIS SEVERITY: A CROSS-SECTIONAL STUDY" is aimed to examine specific blood platelet parameters, such as PLT and LINF counts, MPV and PLR as OA biomarkers, stratifying for OA grade, age, gender, and BMI differences. The work is well structured and the design is scientific. The following comments need to be addressed to improve the quality of the manuscript before acceptance.
1. The introduction should have referred more recent works. Please include work form last 2 years. This is missing in the manuscript.
2. L130- " The KL grading system was used for classifying radiographic OA by two physicians with disagreements resolved by consensus with a third senior physicians" The authors should reveal, what are the disagreements and how they have come to consensus. This should be included in the manuscript, as it would be helpful for future researchers to avoid such errors.
3. binary logistic regression model Overfits on high dimensional data. How did the authors take care to overcome this drawback.
4. Suggest to include graphical representation of the statistical data
5. L283- The authors mentions the limitations of the current work. How does the authors plan to address such limitations int he future work to give more accurate results.
6. Please re-write the conclusions in bullet points, if possible for better readability.
Comments on the Quality of English Language
Minor editing required.
Author Response
- The introduction should have referred more recent works. Please include work form last 2 years. This is missing in the manuscript.
As suggested by the reviewer, we have replaced some older references with more recent ones. However, with regard to the references on platelets, we have kept the existing ones as there are no more recent sources available than those we included.
- L130- " The KL grading system was used for classifying radiographic OA by two physicians with disagreements resolved by consensus with a third senior physicians" The authors should reveal, what are the disagreements and how they have come to consensus. This should be included in the manuscript, as it would be helpful for future researchers to avoid such errors.
Thank you for pointing this out. We agree that detailing the process for resolving disagreements in radiographic OA classification is important for transparency and reproducibility. In our study, disagreements between the two initial physicians were typically related to the interpretation of borderline cases where the degree of joint space narrowing or osteophyte formation was subtle. These instances were discussed in detail with the third senior physician, who provided additional insights based on extensive experience with the KL grading system.
To reach consensus, the three physicians reviewed the radiographic images together, discussing the criteria and the specific features observed. Disagreements were resolved through a collaborative discussion where the senior physician's opinion was pivotal in ensuring that the final classification was consistent with established grading criteria. This process helped to standardize the grading and minimize subjectivity.
We have included a detailed description of this consensus process in the revised manuscript to guide future researchers and ensure clarity on how such disagreements are handled.
- binary logistic regression model Overfits on high dimensional data. How did the authors take care to overcome this drawback.
We thank the reviewer for the comment, which allowed us to realise that in the 'Statistical analysis' section we had not specified that we had used k-fold cross-validation to determine whether there was over-fitting of the model. In the 'Results' section, we have added additional data on the accuracy of the model.
- Suggest to include graphical representation of the statistical data
As suggested, to better understand the variations in hematologic parameters in relation to the severity of osteoarthritis, the data were analyzed using boxplots. These boxplots represent the distribution of PLT, MPV, LINF, and PLR among patients with early OA (KL < 3) and those with moderate/severe OA (KL ≥ 3).
- L283- The authors mentions the limitations of the current work. How does the authors plan to address such limitations int he future work to give more accurate results.
As suggested by the reviewer, we have added a discussion at the end of the discussion section on how to address the limitations of the study in future research
- Please re-write the conclusions in bullet points, if possible for better readability.
As suggested by the reviewer, we added the bullet points after the conclusion paragraph.
Comments on the Quality of English Language. Minor editing required.
We have reviewed the English in the text.
Round 2
Reviewer 1 Report
Comments and Suggestions for Authors
The authors have addressed all comments successfully.
Comments on the Quality of English LanguageMinor.
Reviewer 2 Report
Comments and Suggestions for Authors
The comments were incorporated in the manuscript and may be accepted for publication .
Comments on the Quality of English LanguageMinor editing needed